# Ultrasonography and Infrared Thermography as a Comparative Diagnostic Tool to Clinical Examination to Determine Udder Health in Sows

**DOI:** 10.3390/ani12192713

**Published:** 2022-10-09

**Authors:** Sebastian Spiegel, Florian Spiegel, Matthias Luepke, Michael Wendt, Alexandra von Altrock

**Affiliations:** 1Clinic for Swine, Small Ruminants, Forensic Medicine and Ambulatory Service, University of Veterinary Medicine Hannover, Foundation, 30173 Hannover, Germany; 2Department of General Radiology and Medical Physics, University of Veterinary Medicine Hannover, Foundation, 30173 Hannover, Germany

**Keywords:** imaging, mammary gland, postpartum dysgalactia syndrome

## Abstract

**Simple Summary:**

The udder health of sows is most important to raise healthy piglets. The aim of the study was to investigate a possible advantage of infrared thermography and ultrasonography over the clinical examination of the udder of sows. For this purpose, both clinically healthy sows with inconspicuous udders on palpation before and after birth (n = 35) and sows at the time of weaning (n = 107) were examined. Images of thermography and ultrasound revealed no pathological alterations in the clinically healthy sows. A physiological statistically significant increase in the udder surface temperature and the thickness of the parenchyma during the three weeks ante partum was observed. After weaning, abnormalities in the appearance of roundish nodules of the parenchyma were detected sonographically in 10.3% of the examined sows, while the demonstrated nodules were unrecognised clinically in two out of eleven sows. The changes could also be demonstrated thermographically because of a statistically significant lower surface temperature above the nodules compared to the remaining skin of the mammary gland. However, scratches on the udder skin showed similar temperature changes. Therefore, thermographic images without prior inspection of the udder can lead to misinterpretation.

**Abstract:**

The aim of the study was to examine whether the use of infrared thermography and ultrasonography can complement or replace the clinical examination of the sows’ mammary glands for pathological alterations. Sows of different parities with inconspicuous udders on palpation before and after birth (n = 35) and sows at the time of weaning (n = 107) were examined. Thermal images were taken from both sides of the udder, while ultrasound pictures were taken from four sides of the respective mammary glands. Within three weeks before birth, a statistically significant increase in the average surface temperature of the glands of about 1.54 °C and of the thickness of the parenchyma of about 1.39 cm could be observed. After weaning, in 10.3% of the examined sows, roundish hyperechogenic nodules were detected sonographically in the glands´ parenchyma. The average skin temperature above the nodules was 1.24 °C lower compared to the total skin area of the altered complex. However, scratches on the udder skin showed similar temperature changes. In two sows, the nodules remained undetected during the clinical examination. Therefore, sonography seems to be superior compared to clinical and thermographic investigations, although it proved to be very time-consuming.

## 1. Introduction

Postpartum dysgalactia syndrome (PPDS) represents one of the most economically important diseases in sow herds worldwide [1,2], affecting both the sow and her litter. Mastitis is mentioned as one symptom of the disease complex, although its role as part of the complex is actually questioned [3]. Signs of acute mastitis usually develop before [4]—or more frequently one to two days after—farrowing [5] and are characterised by local inflammation of one or more mammary gland complexes of the udder. Highly altered complexes show swelling, pain, increased temperature, redness and induration [6,7], and can be associated with a disturbed general condition, fever, hypo- or agalactia [8,9,10]. If the infection is not diagnosed and treated at an early stage, chronic processes develop from acute mastitis, characterised by the formation of granulomas or abscesses in the mammary gland [11], and loss of function. Chronic mastitis leads to premature removal of sows with economic impact [12] because of suggested lower milk production in subsequent lactations [11]. In order to avoid economic losses as a consequence of udder alterations, a thorough examination of the mammary glands, especially during the stage of involution, is mandatory [13].

At present, mastitis diagnostic in sows is based on clinical evaluation by inspection and palpation of the udder in addition to the macroscopic and bacteriological control of milk secretion. In recent years, new diagnostic tools have attracted interest, including imaging techniques such as infrared thermography (IRT) and ultrasonography.

IRT is a method for displaying the temperature distribution of object surfaces. For this purpose, the infrared radiation emitted by the surface is measured, and then the temperature of the emitting surfaces is calculated and displayed in colour-coded form as a thermogram or thermal image. These thermal images can be used for detecting localised anomalies [14]. Its application has already been investigated for diagnosing mastitis in cows [15,16] and sheep [17], while the technique was tested in swine to evaluate rectal temperature [18,19] or to predict estrus from vulva surface temperature [20,21]. The use of this method would be highly advantageous in the diagnosis of udder alterations in sows with its quick, non-contact and economic applicability.

In recent years, ultrasonography has become a valuable method for assessing the female reproductive tract in swine [22]. It is widely used for pregnancy diagnosis, and progress has also been made in its use to examine the non-gravid uterus in recent years [23]. In contrast, ultrasonographical investigations of mammary glands of sows do not belong to routine diagnostics, while examination of the bovine udder is becoming more and more common. Paulrud et al. [24] considered ultrasound as well as thermography to be useful methods to evaluate teat tissue integrity in dairy cows.

The aim of the present study was to investigate whether IRT or ultrasonography can improve the early identification of udder alterations in sows. For this purpose, sows were examined clinically ante partum (a.p.) and post partum (p.p.) during lactation as well as after weaning. The findings were compared with the results of thermographic and sonographic surveys.

## 2. Materials and Methods

On each examination day, ambient temperature, air humidity and air velocity were measured (Testo 400, Testo AG, Lenzkirch, Germany) in the respective units before starting the examination of the sows. The measurements were taken at the height of the sows in their immediate vicinity. In order to carry out the investigation under approximately the same conditions, examination took place only during feeding in the morning from September to April.

In Part I of the study, 35 sows (Landrace X Large White) of a farrow-to-feeder herd with about 80 sows were selected. For selection, sows had to be clinical healthy during the observation time and had to have no history of mastitis or PPDS. They were divided into three groups according to the parity (group 1: 1st parity (n = 17), group 2: 2nd/3rd parity (n = 15), group 3: > 3rd parity (n = 3)). The udders of the sows were investigated six times (day 21 a.p., day 7 a.p., day 1 a.p., day 1 p.p., day 3–4 p.p., day 14 p.p.). The mammary glands were identified according to the side of the udder (right side: R, left side: L) and numbered from cranial to caudal.

In Part II of the study, the udders of 107 sows (Landrace X Large White) housed on three different farrow-to-feeder farms (farm A: n = 35, farm B: n = 35, farm C: n = 37) were examined at the time of weaning (about 4 weeks p.p.). Age data or the number of pregnancies of the sows was not given.

All sows were investigated clinically, thermographically and sonographically. The thermographic and sonographic examinations were always carried out by the same person; both performed the clinical examinations of the sows.

Clinical examination of the standing sows took place during feeding in the morning and included rectal temperature, heart and respiratory rate and behaviour. In the farrowing unit, suckling piglets were removed for the time of examination. The udders were dry-cleaned, and contaminants such as dung and straw were removed. All mammary glands were inspected visually and palpatorily. Morphology and inflammation symptoms of the glands were recorded using a scoring system (Table 1). Additionally, scratches and further pathological changes, such as nodules, were recorded.

After clinical examination, thermal images were taken almost at a right angle from both sides of the standing sow´s udder from a distance of about 150 cm and parallel to the ground using a high resolution inspect infrared camera (VarioCAM^®^, Infratec GmbH, Dresden, Germany, 1.0/25 mm standard lens, 384 × 288 pixels, adjusted to the actual room temperature). An emissivity value of 0.96 was preset, which is a good assumption for skin with little hair, according to Diakides et al. [25].

Before the images were taken in the farrowing unit, infrared lamps in creep areas were switched off. The sows had to stand untouched at least three minutes after palpating the udder.

Mammary glands that could not be completely depicted were not included in the analysis. The mean surface temperature of each mammary gland, as well as of the cranial and caudal part of each gland, was calculated using the image analysis software package IRBIS^®^ 3 plus (Infrared Thermographic Software, InfraTec GmbH, Dresden, Germany) and compared with each other. For this, the region of interest was selected manually. In case of clinically apparent alterations of any glands, conspicuous skin areas were analysed separately.

Subsequently, ultrasound examination was performed. From clinical healthy sows, images were taken from the 1st, 3rd, 5th and 7th mammary glands of both udder sides. The investigation of sows at the time of weaning included ultrasound pictures of all mammary glands. A portable ultrasound system with a linear transducer 5–10 MHz (Fazone CB^®^, Fujifilm, Physia GmbH, Neu-Isenburg, Germany) was used. Penetration depths of 5 cm (day 21 a.p., day 7 a.p.) and 8 cm (day 1 a.p., day 1 p.p., days 3–4 p.p., day 14 p.p. and at the time of weaning) were chosen. After applying ultrasound gel, the probe was placed cranial, caudal, medial and lateral of each gland (described by Sporn [26]), and two images were taken in the respective position and stored in an external hard disk. Care was taken to ensure uniform pressure on the transducer. The thickness of the skin, the muscle layer and the parenchyma were measured (ImageJ version 1.47v, Wayne Rasband, National Institutes of Health, Bethesda, Maryland, USA, 2012, public domain), and the texture described (fine-grained, grained, lack of structure). The tissues were further evaluated using a modified score system based on Trasch [27] and Sporn [26] (Table 2) and compared between the different parity groups.

For statistical evaluation, the program SPSS Statistics^®^, Version 22.0 (64 Bit) for Windows (IBM Inc., Armonk, NY, USA) was applied.

The distribution of all continuous values was tested for normality using the Kolmogorov–Smirnov test and the Shapiro–Wilk test. Mean values of the ambient temperature, relative humidity, air velocity, surface temperature of the cranial and caudal part of the single mammary gland, left and right gland of the same position, surface temperatures of skin areas above chronic alterations of mammary glands (nodules) and the remaining skin of the gland were compared by independent t-test and ANOVA. For non-normally distributed data, such as scale values and layer thickness of each tissue and of the surface temperatures of the mammary glands, the Kruskal–Wallis test and Chi-square homogeneity test for independent samples were used. For small data samples, Fisher´s exact test was applied. The Spearman correlation coefficients between udder skin and rectal temperature, as well as udder skin and ambient temperature, were also calculated. The significance level was set at *p* ≤ 0.05.

## 3. Results

### 3.1. Clinically Healthy Sows before and during Lactation

The 35 pre-selected sows did not show any abnormal findings in the clinical examinations. Summarised over all examination days, the mean rectal temperature was 38.39 °C, the heart rate 116.4 beats/min and the respiratory rate was 34.6 breaths/min. All animals showed normal feeding behaviour. The increased respiratory and heart rates were attributed to excitement due to feeding.

Altogether, the udders of 35 sows—with 14.7 mammary glands on average—were investigated on six examination days. In all parity groups, cranial mammary glands (R1–R2, L1–L2; n = 140) showed fewer scratches in the skin (n = 7; 5.0%) than the posterior glands (n = 56; 14.9%) (R3–R8, L3–L8; n = 375). Swellings of the subcutis were determined in particular at day 14 p.p. (n = 37; 7.0%), whereby all mammary glands were equally affected. Statistically, no differences in those clinical alterations were calculated, neither between the glands in various positions nor between the examination days.

#### 3.1.1. Thermography

The mean temperatures in the farrowing unit and gestation unit were similar, while mean relative humidity and mean air velocity showed discrepancies (Table 3). No correlation between the skin temperature of the mammary glands and the ambient temperature existed.

External influences on thermography are shown in Figure 1. Lying on one side caused a decrease in skin temperature in the area of contact with the floor. Humidity and soiling also led to a drop in surface temperature. Therefore, precautions were taken, such as dry-cleaning the udder and waiting at least 3 min after the manual inspection of the standing sow´s udder to avoid misinterpretation.

The analyses of the thermal images of the individual sow taken on the same day revealed no statistical differences in the mean temperature between cranial and caudal areas of single mammary glands, neither between the pairs of the right and left glands nor between the glands of the same row. Therefore, mean surface temperatures were subsumed under the mean udder skin temperature for each examination day.

Thermal images from the udder of one healthy sow taken on the different examination days are presented in Figure 2. Statistically, there were no significant differences in the mean surface temperatures of the udder skin between the three groups of sows referring to the day of pregnancy or lactation. Therefore, the results of all sows of different parity were summarised according to the days of investigation. During parturition and lactation, an increase in udder skin temperature could be observed. A statistically significant rise in temperature of 1.54 °C on average from 33.39 ± 1.70 °C to 34.93 ± 0.89 °C before parturition (day 21 a.p. to day 1 a.p.) and additionally from day 1 a.p. to day 1 p.p. of 1.38 °C to 36.31 ± 0.92 °C was noted. Until the 14th day p.p., the udder skin temperature rose only slightly by 0.40 °C to 36.71 ± 0.90 °C. Rectal temperature and udder skin temperature correlated statistically significantly (r = 0.574), although the most obvious increase in rectal temperature was only around the time of farrowing, with 0.60 °C on average, and both temperature curves ran almost parallel only after birth (Figure 3).

#### 3.1.2. Sonography

In total, 1680 mammary glands were investigated sonographically. Eight images of each gland were evaluated for the six different examination days. There were no statistical differences in the tissue texture between mammary glands of different locations or depending on the stage of pregnancy or lactation. Skin and parenchyma were always well demarcated from the neighbouring tissue (score 1), showing a 50% to < 100% homogeneous structure (score 2) of medium echogenicity (score 2) and a fine-grained texture as well as a neutral acoustic wave behaviour (score 1). Anechoic areas (score 1) of the skin were only found in one mammary gland of a total of 12 sows, while in the parenchyma, anechoic areas were detected regularly. The two glandular systems within each mammary gland could not be differentiated. Single Doppler signals in the parenchyma were found in all parity groups a.p. (score 1), while the number of signals increased (score 2) in the glands of group 1 immediately after birth (day 1 p.p.) and in group 2 at day 14 p.p. Older sows (group 3) had only a few Doppler signals during all examination days. The udder skin showed no Doppler signal (score 0). All mammary glands had an inhomogeneous (score 4) muscle layer of medium echogenicity (score 2) and coarse-grained texture. In most glands (94.7%; n = 1675), the muscle layer was well demarcated from neighbouring tissues (score 1). Anechoic and hyperechoic regions (score 1), as well as hyperechoic lines (score 1), could be determined in the muscle layer, and there were very few Doppler signals (score 1).

Comparing the thickness of different tissues of the mammary glands on single examination days, there were no differences due to their position or between the glands of sows of different parity. Therefore, the mean values of the width of glands tissues of sows on specific days of pregnancy and lactation were calculated and compared. Figure 4 shows an example of the increase in thickness of the mammary parenchyma measured from ultrasonographic images in the period from day 21 a.p. and day 14 p.p.

The thickness of the udder skin increased from 0.18 cm ± 0.04 cm on day 21 a.p. to 0.27 cm ± 0.08 cm on day 1 a.p. (Figure 5a). Subsequently, the layer thickness decreased to 0.24 cm ± 0.05 cm up to day 14 p.p. The width of the muscle layer continuously increased on average by 0.7 cm from 1.37 cm ± 0.39 cm to 2.07 cm ± 0.63 cm at day 14 p.p. over the whole observation period (Figure 5b). The increase in thickness of both tissue layers was not statistically significant. In contrast, the parenchyma increased significantly during ante partum from 1.05 cm ± 0.24 cm on day 21 a.p. to 2.44 cm ± 0.57 cm on day 1 a.p., followed by a lower growth to 2.96 cm ± 0.89 cm by day 14 p.p. (Figure 5c).

### 3.2. Sows at the Time of Weaning

At the time of weaning, 76.4% of the clinically examined mammary glands (n = 1,528) out of a total of 107 sows were still completely developed (score 0), 7.3% showed partial involution (score 1), while 16.2% were completely regressed (score 2). In 33.0% of all examined glands, mild swellings of the subcutis could be palpated (score 1), 24.4% of all mammary glands showed scratches on the udder skin and 7.1% had teat injuries (score 1). In the parenchyma of 13 mammary glands (eight completely developed, three showing partial involution, two complete involution), nodular structures were palpated, whereby two of these alterations were first discovered by ultrasound only and were subsequently confirmed by repeated intensive palpations. In nine sows, only a single gland was affected, while two sows showed nodules in two mammary glands. Of the 11 affected sows, 10 cases were discovered on farm C and only one case on Farm A. All but one of the affected glands were localised in the anterior part of the udder (mammary glands 1–3), while mammary gland 6 was only affected once.

Environmental conditions at the time of weaning varied between the three farms. The discrepancy of the mean relative humidity was more than 10% between Farm B and Farm C; the mean air velocity showed a difference of almost 0.3 m/s between Farm A and Farm B (Table 4). Despite these differences, the mean surface temperatures of the sows´ udders did not differ between the farms. Further analysis of the thermographic images showed that undeveloped or fully regressed mammary glands had a colder surface temperature than completely developed glands. However, the difference was not statistically significant.

#### 3.2.1. Thermography

Skin areas above the palpated nodules had a statistically significant lower mean surface temperature (−1.24 ± 0.69 °C) compared to the total skin area of the mammary gland (Table 5). Nonetheless, no significant differences existed between the temperature of the total surface of the affected gland and the surfaces of clinically unremarkable glands on the same side of the udder (Figure 6). Additionally, scratches caused a comparable drop in temperature, although these were not as clearly distinguishable from the surrounding area as the nodular changes in the udder (Figure 6).

#### 3.2.2. Sonography

Nodular tissue alterations of the parenchyma were visualised by sonography. The nodules were hyperechogenic with small anechoic margins (Figure 7b). Doppler examination revealed no signal inside the alterations (Figure 7c). In contrast to the neutral sound in healthy tissue, an acoustic enhancement of the echo signal was shown distal to such alterations.

## 4. Discussion

The aim of the study was to evaluate the implementation of diagnostic imaging techniques, namely, thermography and ultrasound, as diagnostic tools for udder diseases in sows. In Part I of the study, thermal and ultrasound images of clinical healthy mammary glands of sows at different parities in the course of late pregnancy and lactation were investigated to outline normal findings. Subsequently, in Part II, images were taken from udders of sows during the time of weaning, which were not pre-selected for their health status, and findings were compared with results of the clinical investigation.

McManus et al. [28] gave an overview of the diverse applications of the IRT in animal production, e.g., as an indicator trait to estimate the physiological state of an animal in situations of stress, fertility, welfare, metabolism, health and disease detection. Nonetheless, there are some limiting factors that need to be considered when using IRT, such as hairy skin, metal frames of the crates, lying on the ground [29], moisture or dirt on the skin, the effect of weather conditions, time of feeding and laying and circadian rhythms [30]. The influence of moisture and dirt as well as lying on the cold floor could be pictured and were excluded by taking appropriate measures, such as dry-cleaning the udder and observance of a waiting time before taking images of the standing sow´s udder.

Air temperature and humidity influence skin temperatures through heat exchange [31]. To avoid major fluctuations in these environmental factors, thermal images were always taken during feeding in the morning. At the same time, this procedure should prevent fluctuations in udder surface temperature due to the influence of the circadian rhythm, as described for dairy cows [32]. Ambient temperature, air humidity and velocity showed only slight variations, and heat stress, which might lead to a rise in body temperature [33], could be excluded.

The relation between surface temperature and rectal temperature of swine was investigated several times with different outcomes depending on the position of the selected skin area and different environmental factors [34]. In the present study, the mean rectal temperature of the sows was nearly constant up until day one a.p. and increased about 0.60 °C up to day one p.p. Kelley and Curtis [35] also observed a rise in body temperature of about 0.6 °C during a four-hour period a.p., while Littledike et al. [36] recorded an increase of 1.4 °C in association with parturition. At the same time, a distinct rise in the udder surface temperature could be observed, with a significant increase between day 1 a.p. and day 1 p.p. of 1.38 °C due to increased metabolic activity of parenchymal cells [37] in connection with increased perfusion. The increased activity of the parenchyma was also reflected by the statistically significant increasing thickness of the tissue between day 21 a.p. and day 1 a.p. of about 1.4 cm on average. During the final trimester after conception, the parenchyma mass of each gland increased in response to oestrogen, progesterone, prolactin and relaxin [38]. Although in gilts, differences in the growth rate of individual glands were reported according to location, with glands 3, 4 and 5 achieving the largest tissue mass [39], no differences in the growth rate of the glands could be demonstrated due to their position. Additionally, an influence of the parity number on the mammary gland growth rate could not be detected, while Nielsen et al. [40] reported that the amounts of mammary tissue were larger in multiparous than in primiparous sows.

Though the growth of the mammary gland during lactation is described as substantial [41], the presented data showed that from day 1 p.p. on, the thickness of the parenchyma layer only increased slightly, with 0.26 cm on average up to day 14 p.p. However, various factors such as litter size, nursing intensity and the use of a teat in the previous lactation have an impact on the amount of mammary tissue [42], which were not taken into account in the present study, as the focus was laid on the use of the different technologies as a diagnostic tool in udder diseases of sows.

The development of the echotexture from day 21 a.p. to day 14 p.p. showed only a few changes. The homogeneity of the mammary gland parenchyma was between 50% and < 100% on all examination days independent of the parity group, although variations between 20% and 100% occurred. Sporn [26] described an average of 75% homogeneity of the mammary gland parenchyma in gilts. Variations in homogeneity are caused by anechoic areas, which were reported in the bovine udder and represent vessels or milk sinus. The frequency of these anechoic areas varies according to the milk content in the mammary gland [43,44]. In accordance with the results of a study by Hassan et al. in donkeys [45], Doppler sonography allowed a clear distinction to be made between milk sinus and vessels in sows since vessels showed a pulsed Doppler signal and, in contrast to milk sinus, a different colour pattern.

Our results revealed that the number of Doppler signals increased during lactation, indicating an increased blood flow in the udder parenchyma, although a distinct increase in the surface temperature could not be detected. After weaning, mammary blood flow decreases by 40% within the initial 16 h [30]. Thermal images of the sows´ udder revealed the decrease in temperature to be in connection with the involution of the gland but without statistical proof. Chronically altered complexes are often associated with a regression of the mammary gland parenchyma before the end of lactation, caused by less milk production and consequently reduced suckling frequency by piglets, whereby mammary glands that are not regularly suckled also undergo regression [46]. At the time of weaning, chronic mastitis characterised by the formation of abscesses and granuloma becomes apparent [11]. During clinical investigation, nine sows (8.4%) were detected with palpable nodules in the parenchyma of the mammary glands. Two other affected glands were only clinically diagnosed after ultrasound images visualised those changes. According to the findings of Hulten et al. [11], mostly only one mammary gland was affected (81.8%). Using ultrasound, the nodular changes were distinctly defined in the parenchyma. Alterations consisted of anechoic areas with hyperechogenic content and were found in the mammary gland parenchyma of the affected gland. The conspicuous inhomogeneity of the altered parenchyma suggests that homogeneity is a very good assessment criterion for differentiating healthy and chronically altered parenchyma. In our case, alterations were interpreted as small abscesses resulting from chronic puerperal mastitis, which is mainly caused by coliform bacteria but also by streptococci or staphylococci [47]. Alternatively, actinomycosis has to be taken into account, where granulomas are also found in the gland parenchyma [48].

When assessing the skin temperature above the nodules in the parenchyma by thermography, partial areas were identified with a statistically significant lower surface temperature on average of 1.24 ± 0.69 °C in comparison to the remaining area of the affected mammary gland. The IRT findings were always detected on nodules noticed by palpation and sonography. Therefore, thermography seems to be suitable as a diagnostic tool for detecting chronic processes in the udder of sows because IRT measures the surface temperature in a more objective and sensitive way than palpation [49]. However, moisture, contamination, scratches and scabs have to be considered as further factors, which significantly affect the surface temperature of the udder and might lead to misinterpretation.

## 5. Conclusions

Sonography can be used as a complementary imaging method for the clinical diagnosis of local chronic alterations in the udder of sows, although the examination of the entire udder is very time consuming. On the other hand, the examination of the mammary glands by means of thermography showed no advantage over clinical examination, as recording skin lesions such as scratches or scabs is necessary to avoid misinterpretation of circumscribed differences in skin temperature.

## Figures and Tables

**Figure 1 animals-12-02713-f001:**
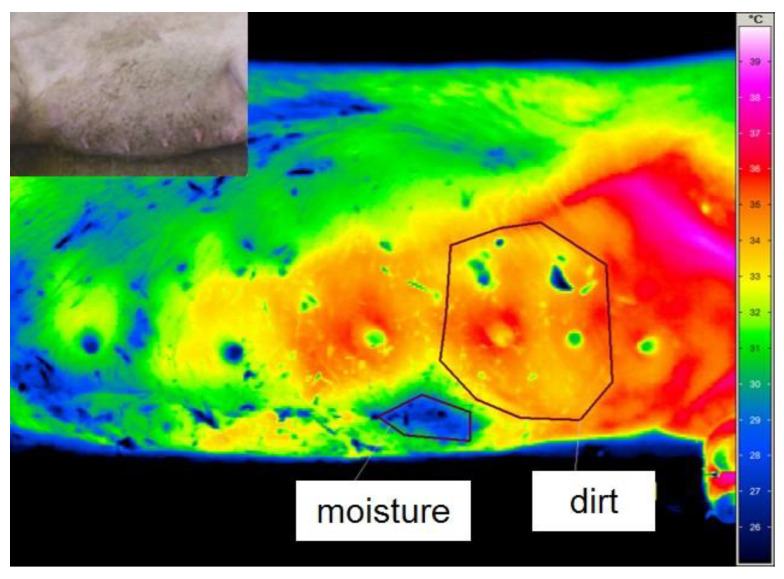
Colour-coded thermal image of the uncleaned udder of a lying sow: marked areas show wet (blue) and soiled (yellow, green and blue spots) areas. The temperature scale is shown on the right side of the figure. The picture at the top left shows the same image of the animal in visible light for better orientation.

**Figure 2 animals-12-02713-f002:**
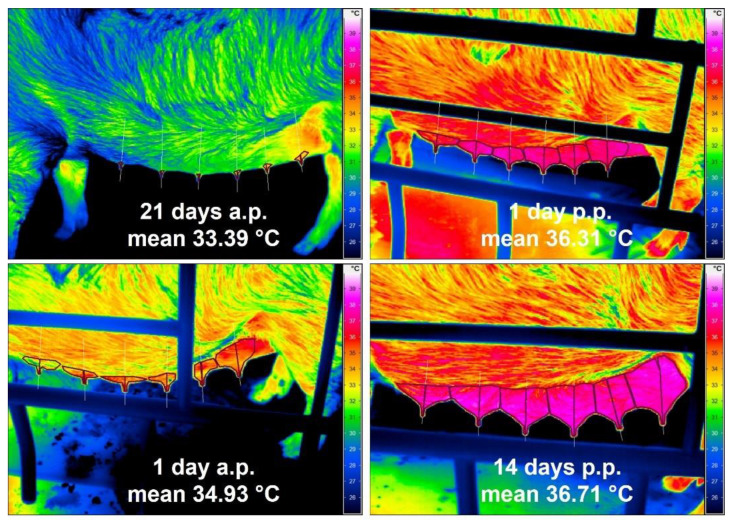
Four thermograms of the left teat row of a sow at four time points (21 d a.p., 1 d a.p., 1 d p.p., 14 d p.p.); mean values represent the mean surface temperature of all mammary glands of all sows (n = 35) on the respective day of pregnancy or lactation.

**Figure 3 animals-12-02713-f003:**
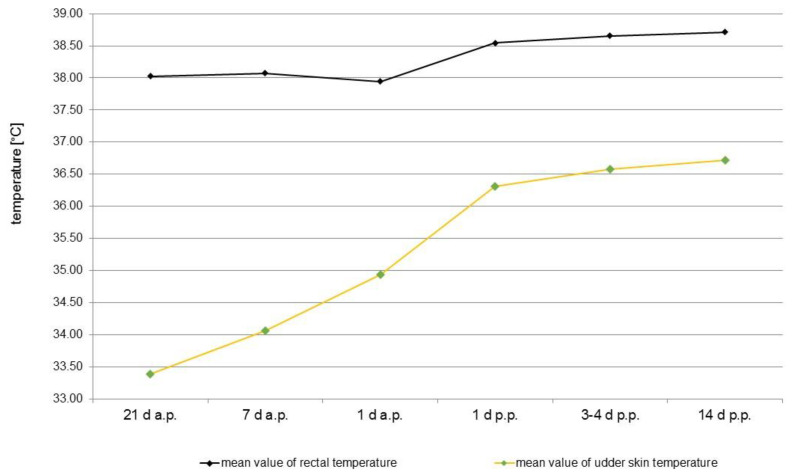
Comparison between the total surface temperature of the udder determined from the mammary glands of all clinical healthy sows (n = 35) and the rectal temperature before and during lactation at different time points (21 d a.p., 7 d a.p., 1 d a.p., 1d p.p., 3–4 d p.p., 14 d p.p.). Mean values of the rectal and surface temperature of the mammary skin are shown.

**Figure 4 animals-12-02713-f004:**
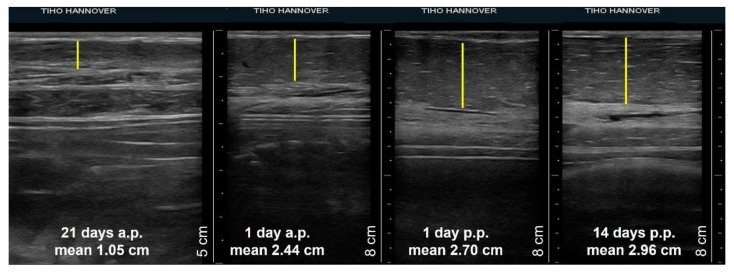
Four sonograms of a sow´s third mammary gland showing the thickness of the parenchyma (yellow line) at day 21 a.p., day 1 a.p., day 1 p.p., day 14 p.p. Mean values represent the mean thickness of the gland parenchyma of all sows during the specific day of pregnancy or lactation.

**Figure 5 animals-12-02713-f005:**
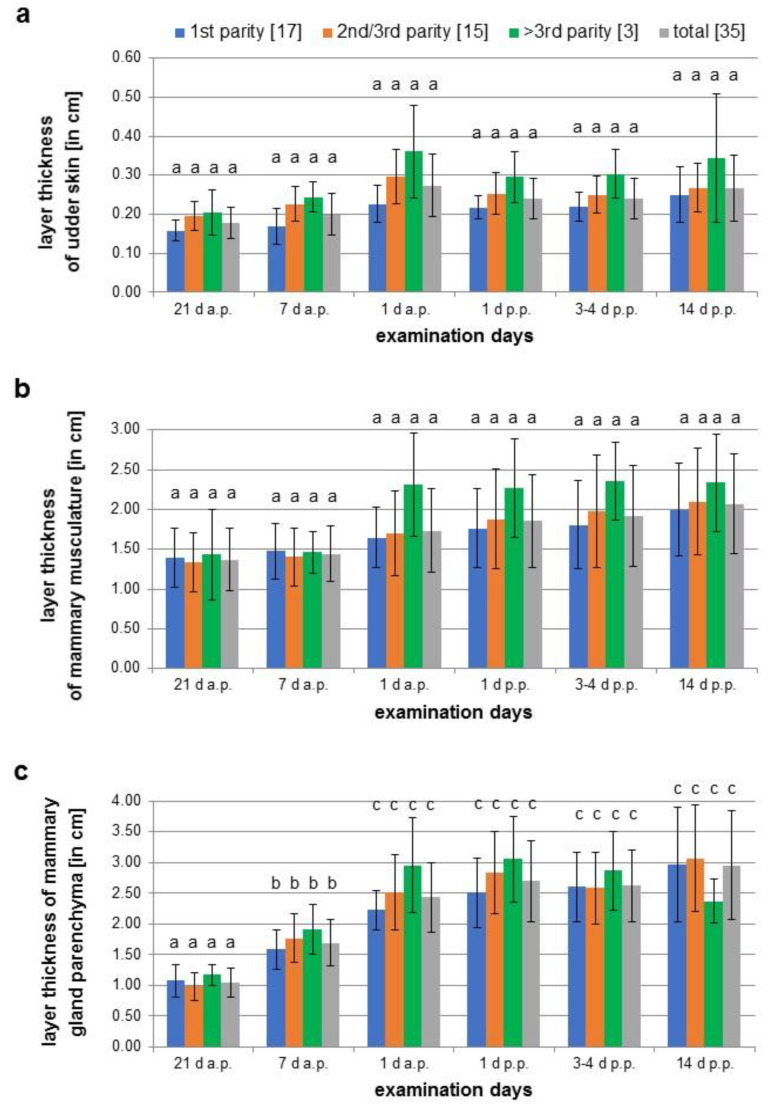
Measurements of the layer thickness of the udder skin (**a**), muscle layer (**b**) and mammary gland parenchyma (**c**) of three sow groups during different parities (1st parity, 2nd/3rd parity, > 3 parity) on six different examination days (21 d a.p., 7 d a.p., 1 d a.p., 1 d p.p., 3–4 d p.p., 14 d p.p.); different letters show significant differences between the age groups and the time of examination (*p* < 0.05).

**Figure 6 animals-12-02713-f006:**
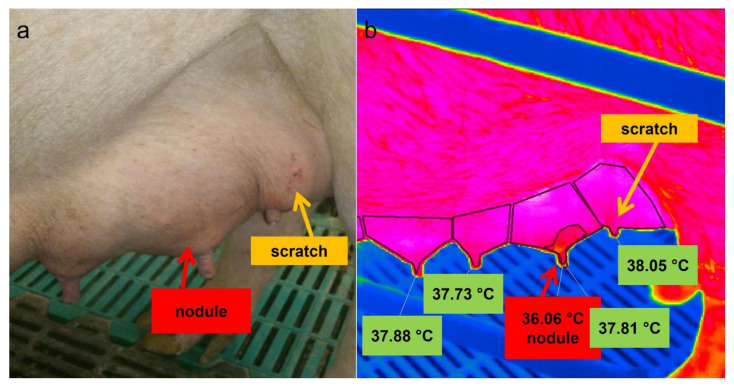
Picture (**a**) and thermal image (**b**) of the left side of a sow´s udder. The mean surface temperature (36.06 °C) of an area above the nodules and of the affected total mammary gland as a whole (37.81 °C) were given as well as the mean skin temperature of the adjacent glands; a scratch is marked by a yellow arrow.

**Figure 7 animals-12-02713-f007:**
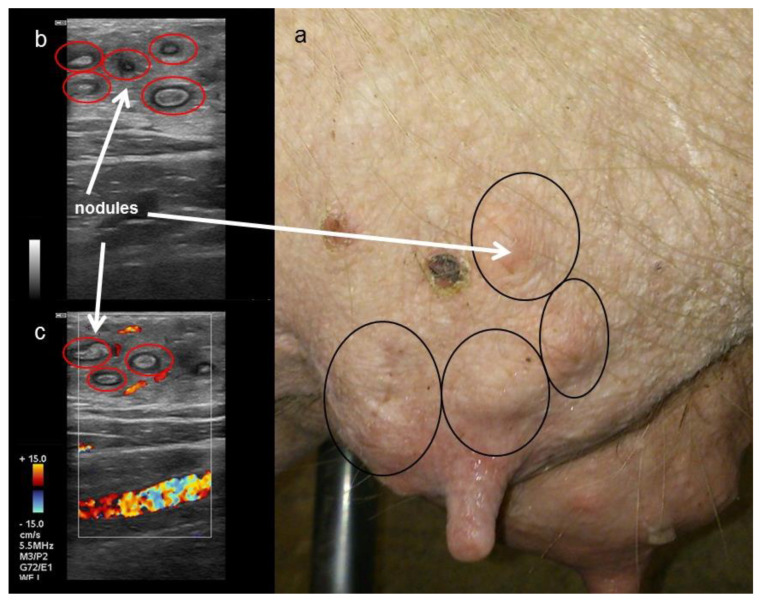
Picture (**a**) and sonographic images (**b**,**c**) of a sow´s mammary gland at weaning. (**b**: B-mode, **c**: Doppler-mode). Red circles mark chronic nodular alterations characterised by (**b**) hyperechogenic centres with small anechoic margins and (**c**) without a Doppler signal. Nodular alterations are circled in black on the right picture (**a**).

**Table 1 animals-12-02713-t001:** Scoring system of individual clinically recorded parameters for assessing the sow udders.

Score for Individual Parameters	Morphology	Swelling/ Redness	Teat Injuries/ Scratches	Nodular Changes in the Parenchyma
0	complete developed	not present	not present	not present
1	partial involution	mild	present	present
2	complete involution	severe	-	-

**Table 2 animals-12-02713-t002:** Score system of individual sonographically recorded parameters for assessing the sow udders (modified from Sporn [26] and Trasch [27]).

Score for Individual Parameters	Homogeneity	Echogenicity	Number of Doppler Signals	Separation from Neighbouring Tissue	Anechoic Areas	Hyperechoic Lines	Acoustic Wave Behaviour
0	-	-	0	-	-	-	
1	100%	high	1–2	distinct	present	present	neutral
2	about 75%	medium	>2	indistinct	not present	not present	acoustic attenuation
3	25% to 75%	low	-	-	-	-	acoustic enhancement
4	<25%	free	-	-	-	-	-

**Table 3 animals-12-02713-t003:** Environmental conditions during the follow-up study of sows of different parity.

Farm	Mean Ambient Temperature (°C)	Mean Relative Humidity (%)	Mean Air Velocity (m/s)
gestation unit (examination on day 21 a.p., day 7 a.p.)	19.44 ± 1.43	62.67 ± 4.95	0.04 ± 0.05
farrowing unit (examination on day 1 a.p., day 1 p.p., day 3–4 p.p., day 14 p.p.)	19.38 ± 1.63	55.23 ± 5.82	0.10 ± 0.12
*p*-value (comparison of the two units)	0.815	0.000	0.000

**Table 4 animals-12-02713-t004:** Environmental conditions in the farrowing units of three different farms during the weaning period. Different superscripts differ statistically significantly (*p* ≤ 0.05) between groups.

Farm	Mean Ambient Temperature (°C)	Mean Relative Humidity (%)	Mean Air Velocity (m/s)
A	22.54 ± 4.36 ^a^	66.98 ± 6.76 ^c^	0.36 ± 0.36 ^e^
B	21.50 ± 1.55 ^a,b^	71.40 ± 5.09 ^c^	0.07 ± 0.10 ^f^
C	20.26 ± 2.37 ^b^	60.44 ± 8.31 ^d^	0.17 ± 0.21 ^f^

**Table 5 animals-12-02713-t005:** Mean surface temperature above the nodules and the total skin area of the affected mammary gland.

Mammary Gland	Mean Surface Temperature above the Nodules (°C)	Mean Surface Temperature of the Affected Mammary Gland (°C)	Difference in Temperature
1	34.73 ± 0.08	36.60 ± 0.57	1.87
2	36.06 ± 0.24	37.81 ± 0.81	1.76
3	35.57 ± 0.18	37.41 ± 0.88	1.84
4	35.31 ± 0.23	36.56 ± 0.70	1.26
5	36.04 ± 0.30	37.41 ± 0.67	1.37
6	37.66 ± 0.09	38.16 ± 0.35	0.50
7	34.92 ± 0.14	36.07 ± 0.59	1.15
8	33.64 ± 0.16	36.36 ± 0.36	2.72
9	34.78 ± 0.11	35.23 ± 0.48	0.45
10	36.68 ± 0.03	37.02 ± 0.45	0.34
11	35.56 ± 0.33	36.58 ± 0.60	1.02
12	35.02 ± 0.43	36.28 ± 0.80	1.26
13	36.01 ± 0.37	36.60 ± 0.48	0.59
mean	35.54 ± 0.21	36.77 ± 0.59	1.24 ± 0.69

## Data Availability

Publicly available datasets were analysed in this study. The data can be found here: Tierärztl Prax Ausg G Grosstiere Nutztiere 2022, 50(01): 30–37, doi:10.1055/a-1696-3952. Additional data are available on request from the corresponding author.

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
