# Peer review of "Ultrasonography and Infrared Thermography as a Comparative Diagnostic Tool to Clinical Examination to Determine Udder Health in Sows"

_animals, 2022, doi:10.3390/ani12192713_

Round 1

Reviewer 1 Report

The main objective of the present study was to investigate the feasibility of using of infrared thermography and ultrasonography compared to the clinical examination to check the health status of sows udder before birth, after birth, and during the weaning period.

The study was well designed and the large number of tested animals is a strength pint for this study.

The findings were well presented, organized, and  provided with illustrations that are easy to understand.

The conclusion was based on the research and was consisent with the methodology.

However, few grammatical and writing errors are to be checked; for instance, (Line 78, Paulrud et al. [24] considered ultrasound as well as thermography as useful ... I think as useful should be were useful)

(Line 326 in the discussion section, The aim of the study was to evaluate the implantation of diagnostic imaging ... the implantation should be replaced with the implementation)

Author Response

Many thanks for the positive review of our manuscript! We have checked the manuscript again for typing errors and inserted your corrections.

However, few grammatical and writing errors are to be checked; for instance, (Line 78, Paulrud et al. [24] considered ultrasound as well as thermography as useful ... I think as useful should be were useful) – Done.

(Line 326 in the discussion section, The aim of the study was to evaluate the implantation of diagnostic imaging ... the implantation should be replaced with the implementation) – Done.

Reviewer 2 Report

Comments to the authors

·       L16: …over the clinical examination..

·       L52: … with the disturbed general conditions …

·       L91-92: … a farrow-to-finish herd with a capacity of about 80 sows

·       L100: ..farrow-to-finish farms

·       L104: provide more details about the record of clinical observations: one or more individuals involved in the clinical examinations

·       L163: clinical examination of the standing sows included rectal temperature, heart and respiratory rate and behaviour. You should report the results in part 3.1

·       L181: Table 3. Environmental conditions during the follow-up study of sows of different parity (mean±SD). You should add in the table p values and superscripts

·       L243: … antepartum …

·       L257: Subsequently, the layer thickness decreased to  0.24 cm ± 0.05 cm was observed up to day 14 p.p

·       L296: Table 4. Environmental conditions in the farrowing units of three different farms during the weaning period. (mean±SD). You should add in the table p values and superscripts

·       L303: Additionally, scratches caused …

Author Response

Thank you for the positive review of our manuscript. Please find the comments on the changes in the text below:

Comments and Suggestions for Authors

  • L16: …over the clinical examination. We inserted “the” into the text.
  • L52: … with the disturbed general conditions … We inserted “a” instead of “the” into the text, because we think it fits better.
  • L91-92: … a farrow-to-finish herd with a capacity of about 80 sows. We don´t think, that “farrow-to-finish herd” is the right term, because the farm included the management of the breeding sows, and piglets until they reach the fattening stage. So, we didn´t change it.
  • L100: ..farrow-to-finish farms  s. above
  • L104: provide more details about the record of clinical observations: one or more individuals involved in the clinical examinations. We added the information as recommended: (L106-L018) The thermographic and sonographic examinations were always carried out by the same person. Both performed the clinical examinations of the sows.
  • L163: clinical examination of the standing sows included rectal temperature, heart and respiratory rate and behaviour. You should report the results in part 3.1

We added the information: L 172-L176: The 35 pre-selected sows did not show any abnormal findings in the clinical examinations. Summarised over all examination days, the mean rectal temperature was 38.39 °C, the heart rate 116.4 beats/min, and the respiratory rate 34.6 breaths/min. All animals showed normal feeding behaviour. The increased respiratory and heart rates were attributed to excitement due to feeding.

  • L181: Table 3. Environmental conditions during the follow-up study of sows of different parity (mean±SD). You should add in the table p values and superscripts

We added the p-values in Table 3.

  • L243: … antepartum …  (here L246) Here we have inserted the previously explained abbreviation (a.p.)
  • L257: Subsequently, the layer thickness decreased to  0.24 cm ± 0.05 cm was observed up to day 14 p.p (here L261): - changed.
  • L296: Table 4. Environmental conditions in the farrowing units of three different farms during the weaning period. (mean±SD). You should add in the table p values and superscripts.

We added the superscripts in Table 4.

  • L303: Additionally, scratches caused … (here L310): - changed

Reviewer 3 Report

This study compares ultrasonography and infrared thermography as methods for examining udder health of sows and examines the value of infrared thermography as a future rapid examination method. The authors found that ultrasonography provides detailed and accurate depiction of abnormalities in the porcine udder, while infrared thermography provides uncertain diagnosis.

The authors have shown that infrared thermography is capable of revealing internal nodules, which can be observed as regional changes in the temperature measured at the surface.

In this study, changes in udder soft tissue and udder surface temperature before and after delivery are shown as basic data for future diagnosis by infrared thermography. These changes are not affected by the number of births, but depend on the number of days before and after delivery.

These studies are concluded to be published as a basic study for future infrared diagnosis of pig udders.

However, the following improvements should be made before publication. After the improvement, this study provides the possibilities of thermographical image analyis on sow's udder.

The required improvements are below.

1. Authors should re-analyzable the area of the sow's udder with multiple nodules, and compile a table with numerical values of the relationship between temperature change and nodules in the ultrasonical image within multiple node-positive sows.

2. Authors should analyse correlation of visible light-surface images and inflared-thermal images, using image processing techniques. This analysis may show that changes at scratches can be distinguished from nodules on inflared images.

3. In the Discussion, the authors should survey the literature on the possibility of machine learning to detect nodules based on infrared images.

4 If machine learning has the potential to improve diagnosis, authors should consider to open image-data including ultrasound, visible light and thermographic of nodule-positive udder and an equal number of control images for comparison, in a Supplement.

Author Response

Thank you for the positive review of our manuscript. You will find our point-by-point response to your comments:

The required improvements are below.

  1. Authors should re-analyzable the area of the sow's udder with multiple nodules, and compile a table with numerical values of the relationship between temperature change and nodules in the ultrasonical image within multiple node-positive sows.

We added Table 5 and inserted the mean surface temperature above the nodules and the total skin area of each affected mammary gland.

  1. Authors should analyse correlation of visible light-surface images and inflared-thermal images, using image processing techniques. This analysis may show that changes at scratches can be distinguished from nodules on inflared images.
  2. In the Discussion, the authors should survey the literature on the possibility of machine learning to detect nodules based on infrared images.
  3. 4 If machine learning has the potential to improve diagnosis, authors should consider to open image-data including ultrasound, visible light and thermographic of nodule-positive udder and an equal number of control images for comparison, in a Supplement.

Many thanks for the very good suggestions for further analysis of the optical and infrared images. In our opinion, however, such an analysis would go beyond the scope of this work, so that it could only be carried out in a second publication. In this work, above all, the procedure for taking the thermal images should be worked out and optimised (constant recording conditions, timing of the recordings, special preparation of the animals). The evaluation of the images with the help of machine learning would certainly be very promising, but would probably require even more images for training. 

Reviewer 4 Report

The study is interesting as the comparison between termography and ultrasonography has been previously analyzed in other species. There is one question that is not clear enough. Authors say state that to carry out the investigation under the same conditions and to reduce the influence of ambient and udder temperatures, examination took place only during feeding in the morning, from September to April. They should explain why, and move the paragraph to Material and Methods (lines from 173 to 176). 

On the other hand, Discussion should be improved as there is practically no mention to other species in which  the usefulness of either termography or ultrasonography has been already analyzed. 

Author Response

Thank you very much for the good evaluation of our manuscript. In the following you will find our point-by-point response to your comments:

The study is interesting as the comparison between termography and ultrasonography has been previously analyzed in other species. There is one question that is not clear enough. Authors say state that to carry out the investigation under the same conditions and to reduce the influence of ambient and udder temperatures, examination took place only during feeding in the morning, from September to April.

They should explain why, and move the paragraph to Material and Methods (lines from 173 to 176):

            The sentence was moved to L 90 (first section of Material and Methods). The reason was discussed (L360 – L366): Air temperature and humidity influence skin temperatures through heat exchange [31]. To avoid major fluctuations in these environmental factors, thermal images were always taken during feeding in the morning. At the same time, this procedure should prevent fluctuations in udder surface temperature due to the influence of the circadian rhythm, as described for dairy cows [32]. Ambient temperature, air humidity and velocity showed only slight variations, and heat stress, which might lead to a rise in body temperature [33], could be excluded.

On the other hand, Discussion should be improved as there is practically no mention to other species in which  the usefulness of either termography or ultrasonography has been already analyzed. 

Thank you very much for this important objection. We have reviewed the manuscript for this and discussed the possibilities of how far further references to the areas would give the reader additional information without lengthening the manuscript and the literature list too much. Areas of application of thermography in other animal species were mentioned in the introduction, whereby the focus was already placed on examinations of the udder of cows and sheep (L 68-70).

Sonography is an important diagnostic tool in various areas of medicine including veterinary medicine. Therefore, the application areas, which we mentioned, were limited to the pig (L73-L75). In addition, the comparative study of thermography and sonography on the udder of cattle by Paulrud et al. was mentioned.

We have therefore decided not to add to this point.

Round 2

Reviewer 2 Report

No comments 

Reviewer 3 Report

I look forward to further research by the authors.